# Deceptive Risk Minimization: Out-of-Distribution Generalization by Deceiving Distribution Shift Detectors

## Abstract

This paper proposes *deception* as a mechanism for out-of-distribution (OOD) generalization: by learning data representations that make training data *appear* independent and identically distributed (iid) to an observer, we can identify stable features that eliminate spurious correlations and generalize to unseen domains. We refer to this principle as *deceptive risk minimization* (DRM) and instantiate it with a practical differentiable objective that simultaneously learns features that eliminate distribution shifts from the perspective of a detector based on conformal martingales while minimizing a task-specific loss. In contrast to domain adaptation or prior invariant representation learning methods, DRM does not require access to test data or a partitioning of training data into a finite number of data-generating domains. We demonstrate the efficacy of DRM on numerical experiments with concept shift and a simulated imitation learning setting with covariate shift in environments that a robot is deployed in.

## 1 Introduction

Is there an unbridgeable gap between in-distribution (ID) and out-of-distribution (OOD) generalization in machine learning? Or can the distinction be erased by a change in perspective? Traditional wisdom holds that there is a vast chasm between the two settings. Applications where training environments are representative of test environments (e.g., via careful curation of large-scale datasets) have seen remarkable empirical progress and real-world impact. This success is backed by a deep theoretical understanding of ID generalization from decades of progress in statistical learning theory (Shalev-Shwartz & Ben-David, 2014). However, in settings where it is challenging to cover all relevant dimensions of variation exhaustively in the training data — a common occurrence in real-world applications such as robotics, healthcare, and cybersecurity — high-capacity models can absorb spurious correlations and fail catastrophically in test settings where these correlations are altered or even reversed (Liu et al., 2021; Sinha et al., 2022; Li et al., 2025; Arjovsky et al., 2019).

In this paper, we take an *observer-centric* viewpoint on the gap between ID and OOD generalization. Our starting point is the following basic observation: from the perspective of an external observer who cannot discern distribution shifts in a sequence of data, *OOD generalization is equivalent to ID generalization*. As an example, consider an observer responsible for overseeing the performance of a robot operating in a warehouse. The robot is presented with a sequence of objects to place into a receptacle, while the observer records the corresponding sequence of bits denoting success (1) or failure (0) of the robot. During this process, the robot encounters changes to its lighting conditions, appearances of objects, and its visual backdrop. However, the robot is able to maintain reliable performance throughout these changes, with only a small-but-consistent failure probability. As a result, the observer is completely oblivious to the distribution shifts faced by the robot: the data recorded by the observer has shed its spurious, domain-specific cues and appears independent and identically distributed (iid). In a sense, the robot has *hidden* the distribution shifts from the observer.

The core idea of this work is to translate this observer-centric perspective on generalization into a prescriptive mechanism for OOD generalization. Suppose that a learner is presented with a sequence of training data that exhibits (potentially mild) distribution shifts. Then, by learning data representations that eliminate these distribution shifts from the perspective of an observer, we can identify stable features that do not rely on spurious correlations and generalize to unseen domains.

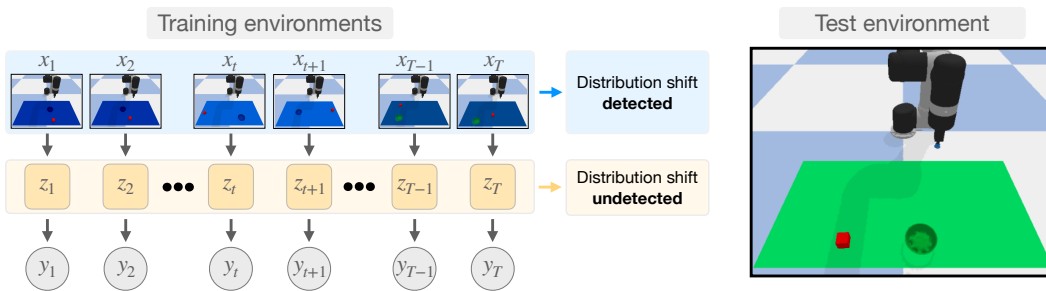

Figure 1: Deceptive risk minimization (DRM): by learning data representations that make training data *appear* iid to an observer (left), we can identify stable features that eliminate spurious correlations and generalize to unseen domains (right). In the figure, the robot's training environments undergo small-but-structured changes in the colors of the table and objects; DRM learns a representation that is insensitive to these changes, which results in generalization to environments with significantly different appearances.

For example, if the robot encounters periodic changes in lighting conditions or visual backdrops (Fig. 1), an encoding of observations that hides these changes from a distribution shift detector will eliminate sensitivity to the changes and result in robust performance when spurious correlations in training data are significantly exaggerated or reversed.

Concretely, we formulate this learning mechanism as an adversarial game (Fig. 1) which we refer to as *deceptive risk minimization* (DRM). An encoder network learns to generate representations that support minimization of a task-specific loss while simultaneously eliminating distribution shifts from the perspective of a detector presented with a sequence of learned representations. We assume that this sequence is presented in the order in which training data were curated, and thus preserves the structure of natural distribution shifts in the original data. For example, in robotics, training data are often collected in environments that vary over time either discretely (e.g., a change in the color of the table) or continuously (e.g., a continuous change in ambient outdoor lighting over a day). We are interested in settings where the training data sequence exhibits some distribution shifts, which are exaggerated or reversed at deployment time. Importantly, unlike prior work in domain adaptation (Ben-David et al., 2010; Ganin et al., 2016; Zhang et al., 2015; Long et al., 2018) or invariant representation learning (Arjovsky et al., 2019; Peters et al., 2016; Ahuja et al., 2020; Krueger et al., 2021), we do not assume access to any data from test environments or that training data are partitioned into different domains corresponding to data-generating distributions; we simply assume that the order of data is preserved. Associating data points with a finite set of domains — either manually or via unsupervised clustering (Le et al., 2025; Murata et al., 2025) — is often impractical or unachievable in settings where there is a continuous change in conditions.

We present a practical instantiation of DRM that utilizes *conformal martingales* (CMs) (Vovk et al., 2022; 2003; Vovk, 2021) for distribution shift detection. CMs offer a general and flexible approach to distribution shift detection, which is often highly effective in practical scenarios (Vovk et al., 2022, Ch. 8). Concretely, the CM approach computes a quantity that remains small when data are iid (or exchangeable), but that can grow quickly in the presence of distribution shifts. We derive an end-to-end differentiable loss that penalizes the conformal martingale computed on encoded inputs; this loss serves to train the encoder to learn representations that eliminate distribution shifts from the perspective of the CM-based detector.

**Summary of contributions.** We introduce deceptive risk minimization (DRM): a novel learning principle that estimates representations that eliminate spurious correlations by deceiving distribution shift detectors. We develop a practical instantiation of DRM via a differentiable loss that penalizes conformal martingales, and demonstrate the efficacy of this representation learning objective in different empirical settings involving covariate and concept shift. Conceptually, DRM creates a bridge between distribution shift detection and OOD generalization, which we hope future work can build on to unlock practical methods for OOD generalization in real-world applications.

## 2 PRELUDE: RANDOMNESS AND STRUCTURE IN THE EYE OF THE BEHOLDER

We provide an illustrative example below in order to explain the key intuitions behind our approach. Consider an imitation learning setting where a human has provided a sequence of examples to a robot demonstrating how to perform a given task (Fig. 1). These demonstrations are provided in

environments that exhibit a small amount of distribution shift. Specifically, the color of the table is varied *slightly* in a structured way: a third of the demonstrations are provided with one table-bowl color combination, the next third with a slightly different color combination, and the final third with another. The standard approach to learning a policy in such a setting is empirical risk minimization (ERM): *assume* that data are iid and learn a mapping from the robot's observations to actions by minimizing a behavior cloning loss on training data. Such a policy performs well when deployed with table colors similar to ones seen during training, but fails with colors that are significantly different (see Sec. 5.3 for numerical results).

The starting point for our approach is the observation that the non-iid nature of data in this setting can be *inferred from the sequence of training environments*. Fig. 2 shows the output from a distribution shift detector based on conformal martingales (described formally in Sec. 4) computed on observations from the training environments. This detector spikes strongly once the table color is changed. Crucially, the detector also spikes when provided with the sequence of latent features from the policy computed via ERM, indicating that the policy's latent representation encodes color information.

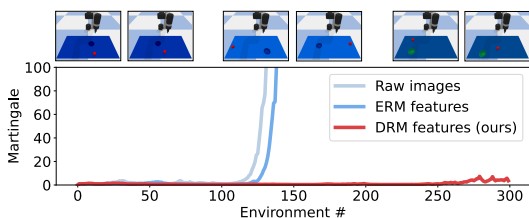

Figure 2: Conformal martingales rapidly detect the distribution shift with raw images or features computed via ERM; in contrast, features from DRM appear iid.

Now consider a policy that eliminates the distribution shift from the perspective of an observer who is only presented with latent policy representations. Intuitively, the data can be made to "appear iid" by eliminating sensitivity to the table color, which leads to OOD generalization to different colors.

## 3 PROBLEM FORMULATION

**Training data.** A learner is presented with a *sequence* of training data $((x_t, y_t))_{t=1}^T \in (\mathcal{X} \times \mathcal{Y})^T$ consisting of input-label pairs sampled from a sequence of random variables $((X_t, Y_t))_{t=1}^T$, which may be dependent and non-identically distributed. This sequential collection of data is a core assumption of our work and departs from the standard practice of shuffling data. If some of the data collection is parallelized (e.g., by multiple human operators collecting data on different robots on the same day), we assume that this data is serialized by committing to a particular ordering.

**Hypothesis, loss, and OOD generalization.** Given the training data, the learner produces a hypothesis $h : \mathcal{X} \to \mathcal{Y}$, which maps inputs to predicted labels. The hypothesis is evaluated according to a loss function $l : \mathcal{Y} \times \mathcal{Y} \to \mathbb{R}$, which compares predicted labels with ground-truth labels. The learned hypothesis $h$ is deployed on a sequence of test data drawn from random variables $((X_\tau, Y_\tau))_{\tau=T+1}^{T+T'}$. The overall quality of the hypothesis is measured by its expected loss on test data $\frac{1}{T'} \sum_{\tau=T+1}^{T+T'} \mathbb{E}\left[l(h(X_\tau), Y_\tau)\right]$. When the random variables $(X_t, Y_t)_{t=1}^{T+T'}$ are iid, this reduces to the standard statistical learning setting. Of interest to us is the case where training data reflect some (possibly mild) distribution shifts, which are significantly exaggerated or reversed for test data.

**Covariate and concept shifts.** We consider two types of distribution shifts: (i) covariate shift, where the distribution of inputs changes over time (e.g., changes in the color of the robot's table in the example from Sec. 2), and (ii) (anti-causal) concept shift, where the conditional distribution of $X|Y$ changes over time (e.g., an image classification setting where the appearances of images for a given label change over time). We discuss potential extensions (e.g., causal concept shift) in Sec. 7.

## 4 ALGORITHMIC IMPLEMENTATION: DECEPTIVE RISK MINIMIZATION

Our goal is to find features that eliminate the distribution shift between training and test settings. Since the learner is only provided with the training sequence, we utilize distribution shifts observed in this data as a proxy. Specifically, we learn features that are stable along the training data sequence — in the sense that they appear iid to an observer — while also minimizing a task-specific loss. We formalize this objective by defining an observer in the form of a distribution shift detector.

**Defining the observer.** Let $(X_1, X_2, \dots)$ be a sequence of input random variables, and let $\phi$ be a mapping from an input $x$ to an encoding $\phi(x) \in \mathbb{R}^{n_d}$. We define an observer $\Delta$ who takes as

input a realization $(\phi(x_1), \phi(x_2), \dots)$ from the random variables $(\phi(X_1), \phi(X_2), \dots)$ and outputs a boolean $\delta \in \{\text{True}, \text{False}\}$ indicating if a deviation from the iid hypothesis has been detected.

**Definition 1** (Practically iid). *A sequence $(\phi(x_1), \phi(x_2), \dots)$ is $\Delta$-practically iid if a distribution shift detector $\Delta$ does not trigger[1] when provided with this sequence as input, i.e., $\Delta((\phi(x_1), \phi(x_2), \dots)) = $ False.*

**False alarm rate (FAR).** For a detector $\Delta$ to be useful, it should not trigger too often when presented with data drawn from a sequence of iid random variables. This is captured by the false alarm rate (FAR), which is the worst-case probability of detection when data are drawn from an iid sequence of random variables (Vovk et al., 2022, Ch. 8).

**Formalizing the DRM objective.** Next, we formulate the objective of deceptive risk minimization (DRM) as a constrained optimization problem. We consider hypotheses $h : x \mapsto \phi(x) \mapsto f(\phi(x)) \in \mathcal{Y}$, which encode inputs using $\phi$ and map these to labels via $f$. We use $(\phi(x_t)|y)_{t=1}^T$ to denote the subsequence of $(\phi(x_t))_{t=1}^T$ with labels equal to $y$. The following optimization problem minimizes the task-specific loss $l$ while searching for a representation that makes the training data $\Delta$-practically iid. We consider two types of constraints aimed at tackling covariate shift and concept shift respectively.

---

**Deceptive risk minimization (DRM)**

$$\inf_{f,\phi} \ \frac{1}{T} \sum_{t=1}^T l(x_t, f(\phi(x_t)))$$

$$\text{s.t.} \ (\phi(x_t))_{t=1}^T \text{ is } \Delta\text{-practically iid} \quad [\text{covariate shift}] \tag{1}$$

$$(\phi(x_t)|y)_{t=1}^T \text{ is } \Delta\text{-practically iid}, \forall y \in \mathcal{Y} \quad [\text{concept shift}]. \tag{2}$$

---

**Instantiation with conformal martingales.** We now describe a particular distribution shift detector $\Delta$ based on conformal martingales (CMs). This will allow us to flexibly handle both covariate and concept shifts. In addition, this detector will allow us to formulate a *differentiable* surrogate for the constraints equation 1 and equation 2 in the DRM optimization problem. Intuitively, the CM approach constructs a quantity that grows quickly when random variables are not iid, and remains small otherwise. In order to detect covariate shift, we first assess how well every encoded data point $\phi(x_i)$ (with $i \leq t$) *conforms* to the sequence of data points $(\phi(x_j))_{j=1}^t$ observed up to time $t$ using a conformity score:

$$\alpha_i^{\text{covariate}} := \min_{j \in \{1, \dots, t\}: j \neq i} d(\phi(x_i), \phi(x_j)), \tag{3}$$

where $d : \mathbb{R}^{n_d} \times \mathbb{R}^{n_d} \to \mathbb{R}^+$ captures how different two encoded data points $\phi(x_i)$ and $\phi(x_j)$ are. In our numerical experiments, we will use the cosine distance or its sharpened form (Ahmad & Mazzara, 2024). In cases where we are interested in detecting concept shift rather than covariate shift, we will alternately use a label-conditioned conformity score:

$$\alpha_i^{\text{concept}} := \min_{j \in \{1, \dots, t\}: j \neq i, y_j = y_i} d(\phi(x_i), \phi(x_j)). \tag{4}$$

The conformity scores are used to compute conformal p-values for each $t \leq T$:

$$p_t^{\text{covariate}} := \frac{|\{i | 1 \leq i \leq t, \alpha_i < \alpha_t\}| + \xi_t |\{i | 1 \leq i \leq t, \alpha_i = \alpha_t\}|}{t}, \tag{5}$$

$$p_t^{\text{concept}} := \frac{|\{i | 1 \leq i \leq t, \alpha_i < \alpha_t, y_i = y_t\}| + \xi_t |\{i | 1 \leq i \leq t, \alpha_i = \alpha_t, y_i = y_t\}|}{|\{i | 1 \leq i \leq t, y_i = y_t\}|}, \tag{6}$$

where $\xi_t \in [0, 1]$ is sampled independently from the uniform distribution on $[0, 1]$.

As shown by Vovk et al. (2022, Ch. 2), the p-values $p_t^{\text{covariate}}$ and $p_t^{\text{concept}}$ are independent and uniformly distributed in $[0, 1]$ in the *absence* of covariate and concept shift respectively. Thus, the CM approach constructs a quantity that measures how far away from being uniformly and independently distributed the p-values are. This is achieved using a betting martingale (Vovk et al., 2022, Ch. 8), whose computation is shown in Algorithm 1.

---

[1]To keep this definition simple, we restrict attention to deterministic detectors (e.g., constructed by taking a detector that has randomness and fixing the seed).

Intuitively, the betting martingale represents the capital of a bettor who gambles *against* the hypothesis that random variables are iid. Large values of the betting martingale $S_t$ thus serve as an indicator of distribution shift. Conversely, in the *absence* of distribution shift, $S_t$ is guaranteed to remain small with high probability. In particular, a detector $\Delta^{\text{CM}}$ which is triggered if the betting martingale ever exceeds a threshold $1/\alpha$ has a false alarm rate bounded by $\alpha$ Vovk et al. (2022).

Conformal martingales suggest a practical method to instantiate the DRM optimization problem: replace the hard constraints in Eq. 1 and Eq. 2 with soft constraints that penalize large values of the betting martingale. The only remaining hurdle is to make the martingale computation differentiable. The steps in Algorithm 1 are differentiable, and hence the only sources of non-differentiability are in the computation of the conformity scores $\alpha_i$ (Eq. 3 or Eq. 4) and the p-values $p_t$ (Eq. 5

---

**Algorithm 1: Betting martingale**

1: **Inputs:** $p_1, \ldots, p_T$; **Outputs:** $S_1, \ldots, S_T$
2: Define $E = \{-1, -0.5, 0, 0.5, 1\}$, $\mu = 0.005$
3: Initialize $C \leftarrow 1$, $C_e \leftarrow 1/|E|$
4: **for** $t = 1$ to $T$ **do**
5:     **for** each $e \in E$ **do**: $C_e \leftarrow (1-\mu)C_e + (\mu/|E|)C$
6:     **end for**
7:     **for** each $e \in E$ **do**: $C_e \leftarrow C_e \left[1 + e(p_t - 0.5)\right]$
8:     **end for**
9:     $C \leftarrow \sum_{e \in E} C_e$;   $S_t \leftarrow C$
10: **end for**

---

or Eq. 6). We follow a procedure similar to Stutz et al. (2021), which differentiates through the calibration procedure of conformal prediction. We replace the minimization operation in Eq. 3 or Eq. 4 by the standard soft-min operation. The computation of $\{i | 1 \leq i \leq t, \alpha_i < \alpha_t\}$ (or $\{i | 1 \leq i \leq t, \alpha_i < \alpha_t, y_i = y_t\}$) is equivalent to the computation of a quantile. This can be approximated by smoothed sorting methods (Blondel et al., 2020; Cuturi et al., 2019), which have a "dispersion" hyperparameter $\sigma$ such that smooth sorting approaches hard sorting as $\sigma \to 0$.

We thus formulate the practical instantiation of DRM as follows by optimizing a weighted combination of the task-specific supervised learning loss (e.g., cross-entropy) and the soft martingale values. We discuss additional algorithmic implementation details in Appendix A.

**Deceptive risk minimization: differentiable objective**

$$\inf_{f,\phi} \quad \frac{1}{T}\sum_{t=1}^{T} l(x_t, f(\phi(x_t))) + \lambda \frac{1}{T}\sum_{t=1}^{T} \tilde{S}_t(\phi(x_1), \ldots, \phi(x_t)), \tag{7}$$

where $\tilde{S}_t(\phi(x_1), \ldots, \phi(x_t))$ is the soft martingale (see Algorithm 2 for details).     (8)

## 5 EXPERIMENTS

We evaluate DRM in three sets of experiments, which seek to investigate the following questions: (1) How effective is DRM in enabling OOD generalization with concept shift or covariate shift and spurious correlations in the training data? (2) Can DRM match the performance of invariant risk minimization (IRM) (Arjovsky et al., 2019), which assumes an oracle partitioning of training data into a finite number of "environments" corresponding to different data distributions? (3) How effective is the conformal martingale approach for distribution shift detection, which forms the bedrock of DRM's algorithmic implementation? Hyperparameters for experiments are listed in Appendix B.

### 5.1 CONCEPT SHIFT: TOY 2D EXAMPLE

We begin with a binary classification task with 2D inputs, where one input dimension correlates strongly but spuriously with the label. Empirical risk minimization (ERM) latches on to this correlation and relies heavily on the spuriously correlated input dimension. However, when the correlation is reversed at test time, the performance of the ERM classifier collapses. This is a 2D version of Colored-MNIST (Arjovsky et al., 2019), which allows for easy visualization.

**Training and test distributions.** The learner is presented with a sequence of training data $(x_t, y_t)_{t=1}^{T}$, where $x_t := [x_t^{[1]}, x_t^{[2]}]$ is two-dimensional and $y_t \in \{0, 1\}$. The first input dimension $x_t^{[1]}$ is drawn from a normal distribution $\mathcal{N}(0, 2^2)$, and a preliminary label $\tilde{y}_t$ is assigned purely as a function of $x_t^{[1]}$: $\tilde{y}_t = 1$ if and only if $x_t^{[1]} \geq 0$. The final label $y_t$ is assigned by flipping $\tilde{y}_t$ with probability 0.25. The second dimension $x_t^{[2]}$ of the input is constructed so that it strongly correlates with the label. Specifically, we first construct $\tilde{x}_t^{[2]} = y_t^{\text{sign}} + u \cdot y_t^{\text{sign}}$, where $y_t^{\text{sign}} = 2y_t - 1$ and $u$ is sampled

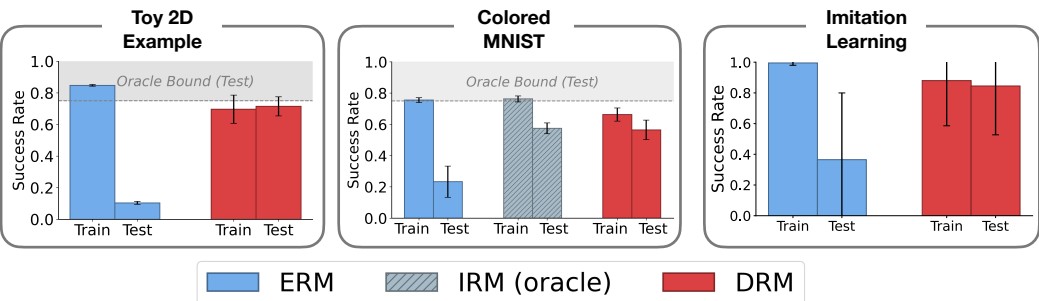

Figure 3: Training and test success rates for the three examples. ERM latches on to spurious correlations or distractors in each case, resulting in severe performance degradation at test time. In contrast, DRM learns stable features that lead to strong generalization. For Colored-MNIST, DRM also matches the performance of IRM, which assumes oracle knowledge of the time at which a distribution shift occurs. For the 2D example and Colored-MNIST, the maximal achievable test success for any robust classifier is 0.75 ("Oracle Bound (Test)").

from the uniform distribution on $[0, 1]$. The learner observes $x_t^{[2]}$, which flips the sign of $\tilde{x}_t^{[2]}$ with a probability that varies smoothly from $p_1$ to $p_T$ over time: $p_t = p_1 + (p_T - p_1)(t-1)/(T-1)$, for $t \in \{1, \ldots, T\}$, with $p_1 = 0$ and $p_T = 0.3$. This time-varying probability is the source of concept shift in the training data, where the distribution of the input conditioned on the label varies slightly over time. At test time, the correlation between $x_t^{[2]}$ and the label $y_t$ is reversed by choosing a flipping probability $p_{\text{test}} = 0.9$. We highlight that IRM (Arjovsky et al., 2019) and its variants (Wang et al., 2022; Krueger et al., 2021; Ahuja et al., 2020; Lu et al., 2021) — which assume that data are separated according to finitely many data-generating distributions — are not directly applicable here since the distribution changes *continuously* for the training data.

**Results.** We train a multi-layer perceptron using both ERM and DRM, and utilize the last hidden representation (unit-normalized) as our feature $\phi(x_t)$ for computing the conformal test martingale in DRM. Fig. 3 (left) compares the performance of ERM with DRM on training and test data (across 10 seeds). The reversal of the spurious correlation results in a dramatic drop in performance on test data for ERM. In contrast, the performance of the classifier learned by DRM is almost entirely unimpacted. This performance also nearly matches an oracle that relies exclusively on $x_t^{[1]}$ for classification, which has a 0.75 classification accuracy on test data.

**Visualizing classifiers.** Fig. 4 visualizes the classifiers learned by ERM and DRM. ERM learns a classifier that heavily exploits the spuriously correlated input dimension $x_t^{[2]}$ in order to maximize training performance, which leads to a collapse in performance on the test distribution. In contrast, DRM learns a classifier that relies almost exclusively on the robust input dimension $x_t^{[1]}$.

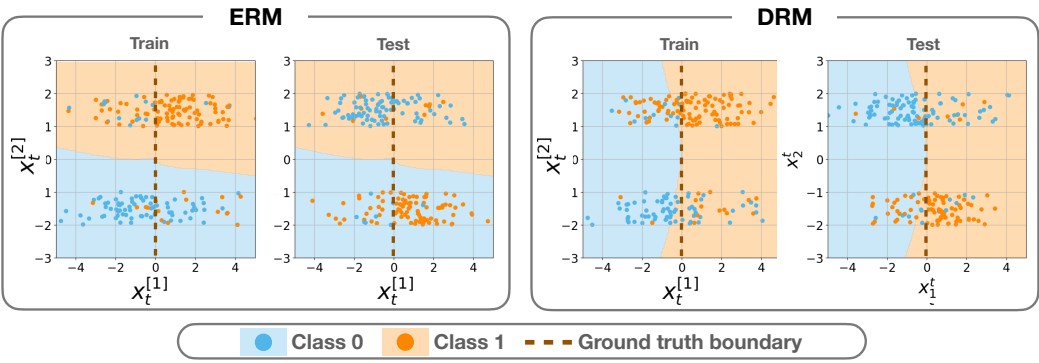

Figure 4: Classifiers learned by ERM and DRM (2D problem). ERM separates data according to the spurious input dimension $x_t^{[2]}$, which leads to poor performance at test time. DRM disregards $x_t^{[2]}$ almost entirely and learns a classifier that is close to the ground truth ($x_t^{[1]} \geq 0$), leading to strong generalization.

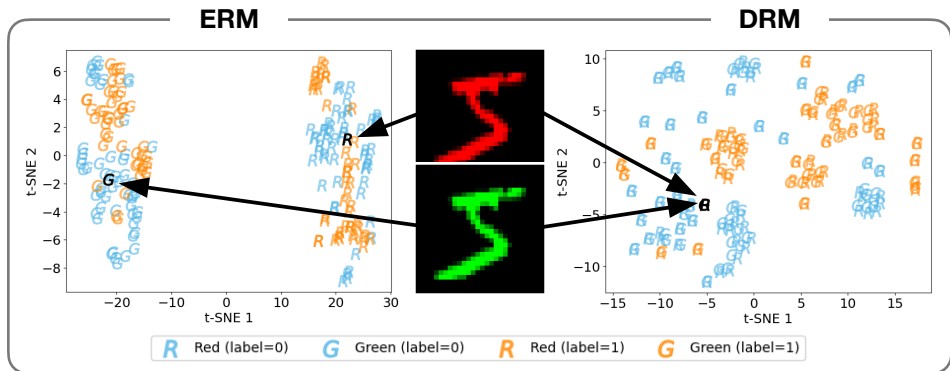

Figure 5: The t-SNE embeddings for features learned by ERM and DRM for Colored-MNIST. ERM embeddings are clustered distinctly by color (R/G). In contrast, DRM embeddings are clustered based on the label, suggesting that DRM has learned to ignore the spurious color information.

## 5.2 CONCEPT SHIFT: COLORED-MNIST

Next, we consider the Colored-MNIST task introduced in (Arjovsky et al., 2019). The goal is to classify MNIST (LeCun et al., 1995) images, where the digits have been colored either red or green. Similar to the toy 2D example, the color is assigned in a way that has a strong (but spurious) correlation with the label. As a result, ERM-based methods that only rely on minimizing training loss exploit the color information to make predictions; when the correlation between color and the label is reversed at test time, performance collapses.

**Training and test distributions.** Each image is first assigned a preliminary label $\tilde{y} = 0$ for digits $0 - 4$ and $\tilde{y} = 1$ for $5 - 9$. The final label $y$ flips $\tilde{y}$ with probability 0.25. A color id $c \in \{0, 1\}$ is obtained by flipping $y$ with probability $p_t$, and the image is colored red if $c = 1$ and green if $c = 0$. The training data sequence consists of examples drawn from two distributions, with the change-point occurring halfway through the data. Specifically, $p_t = 0.1$ for the first half of the data ($t \leq \lceil T/2 \rceil$) and $p_t = 0.4$ for the second half ($t > \lceil T/2 \rceil$). At test time, the probability is chosen to be $p_{\text{test}} = 0.9$.

**Results.** We train a convolutional network with four layers, and use the (unit-normalized) output of the second layer as our feature $\phi(x_t)$ for computing the DRM martingale penalty. Fig. 3 (middle) compares ERM and DRM on the training and test distributions. We also present the performance of IRM, which assumes *oracle knowledge* of the specific point in the training data at which the distribution shift occurs. The reversal of the correlation between the label and the color leads to a significant degradation of performance for ERM. In contrast, DRM achieves a performance that is very similar to IRM, without requiring the training data to be separated into different domains.

**Visualizing features.** In order to obtain more insight into the representations learned by ERM and DRM, Fig. 5 visualizes the features $\phi^{\text{ERM}}(x_t)$ and $\phi^{\text{DRM}}(x_t)$ using their t-SNE embeddings (Maaten & Hinton, 2008). The embeddings are labeled according to the ground-truth labels (blue: 0, orange: 1) for the corresponding input images, along with the color (red: R or green: G) that was applied to the image. The ERM embeddings form two distinct clusters corresponding to the *color* of the image, confirming that ERM learns to rely almost exclusively on the color rather than the shape of the digit. In contrast, the DRM embeddings are separated based on the label rather than the color. The figure shows a grayscale image colored red or green; these images are mapped to an almost identical t-SNE embedding by DRM, suggesting that DRM has learned to ignore the spurious color information.

## 5.3 COVARIATE SHIFT: IMITATION LEARNING

**Training and test distributions.** For our final example, we consider the imitation learning setting from Fig. 1, which involves covariate shift across environments that the robot is trained and deployed in. The task is to pick up and place a red block into a bowl using observations from an RGB camera. The training data consists of 300 expert demonstrations of pick-and-place locations, which are provided in different environments. A third of the demonstrations are provided with one table-and-bowl color combination, the next third with a slightly different combination, and the final third with another combination; these are visualized in Fig. 1. At test time, the bowl and table background color are changed to a novel combination that significantly exaggerates the variation in green and blue channels seen during training (Fig. 1 right); see Appendix D for RGB values.

**Policy training.** We utilize the transporter network approach (Zeng et al., 2021), which uses two separate neural networks for picking and placing objects. For simplicity, we adopt the same network architecture for both picking and placing the red block (instead of the key-query placing model in (Zeng et al., 2021)). Each model takes RGB image observations as input. We use residual networks (ResNets) (He et al., 2016) with 36 total layers (convolutional and residual) that form an hourglass encoder-decoder structure. The models are trained to output an image that predicts per-pixel values corresponding to a likelihood that the robot should move to that location for picking / placing. The pick and place models are both trained via a supervised objective in the form of the cross-entropy loss between predicted and demonstrated pick / place locations. We find that the picking network is not impacted by the distribution shifts in table and bowl colors (since it is trained to locate the red block, whose color does not change). As a result, we only apply the DRM objective to the placing network.

**Efficacy of detector.** Fig. 2 visualizes the martingale values on the training data sequence computed using raw image observations, the features learned via ERM, and the features learned via DRM. As the figure illustrates, the conformal martingale is highly sensitive even to the mild distribution shift that occurs between the first 100 and seconds 100 environments. The martingale value spikes rapidly after the distribution shift for both the raw images and the ERM features. In contrast, the DRM features successfully eliminate the distribution shift from the perspective of the CM.

**Results.** As shown in Fig. 3 (right), DRM learns a policy that is robust to the distribution shift observed between training and testing. In contrast, the near-perfect training performance of a pure behavior cloning objective (ERM) degrades significantly for test environments.

# 6  RELATED WORK

**Distribution shift detection.** Traditional methods for distribution shift detection use batch-based statistical hypothesis testing in order to conclude if a distribution shift has occurred between training and test data (Gretton et al., 2012; Rabanser et al., 2019; Kulinski et al., 2020; Farid et al., 2024). In contrast, DRM relies on *online* methods for distribution shift detection, which have been developed relatively recently. These methods are provided with a stream of data, with no demarcation of where a distribution shift may have occurred. In addition to conformal martingales, methods include universal inference (Ramdas et al., 2022), e-processes (Shin et al., 2022), and recency prediction (Luo et al., 2024; Saha & Ramdas, 2024). Theoretical work has characterized the efficiency with which various methods detect distribution shifts (Shin et al., 2022; Ramdas et al., 2022). Our work creates a bridge between the problem of *detecting* distribution shifts and that of *generalizing* to distribution shifts.

**Domain generalization, invariance, and causality.** Our work is closely related to invariant risk minimization (IRM) (Arjovsky et al., 2019), and the significant amount of subsequent work that it inspired (see, e.g., (Wang et al., 2022; Krueger et al., 2021; Ahuja et al., 2020; Lu et al., 2021)). IRM and its variants seek to find representations that underlie *causal* mechanisms (Schölkopf et al., 2021; Peters et al., 2016; 2017) that generate data. This objective is typically approximated via different regularization schemes (Arjovsky et al., 2019), distributionally robust optimization (Krueger et al., 2021), or via game-theoretic training methods (Ahuja et al., 2020). Practically, the key distinction between IRM and DRM is that we do not assume that data points are associated — either manually or via unsupervised clustering (Le et al., 2025; Murata et al., 2025) — with a finite number of data-generating distributions. This assumption is often impractical or not faithful to reality, e.g., in robotics settings where distribution shifts occur *continuously* as data is being collected (Sinha et al., 2022). Our numerical experiments in Section 5.2 show that DRM can achieve similar performance to IRM without oracular knowledge of distribution shift times. Conceptually, DRM provides a different mechanism for OOD generalization built on the idea of deceiving distribution shift detectors.

**Domain adaptation and online adaptation.** The objective of aligning training and test distributions also underlies domain adaptation methods, e.g., techniques that align features for training and test distributions (Ben-David et al., 2010; Ganin et al., 2016; Ganin & Lempitsky, 2015; Zhang et al., 2015; Long et al., 2018; Gong et al., 2016; Li et al., 2018; Courty et al., 2016), or ones that re-weight training data points to match the test distribution (Shimodaira, 2000; Huang et al., 2006; Lipton et al., 2018). Domain adaptation methods typically assume that labeled "source" data points are separately identified from unlabeled or sparsely labeled "target" data points that come from the test distribution. In contrast, DRM does not assume that data are separated into different sources. Similar to domain generalization methods (e.g., IRM or its variants), we also do not assume access to data from the particular test distribution of interest.

## 7 DISCUSSION AND FUTURE WORK

We have introduced *deceptive risk minimization* (DRM): a novel learning objective aimed at identifying stable features that eliminate spurious correlations by hiding distribution shifts from an observer. Our practical instantiation augments a standard ERM loss with a differentiable objective based on conformal martingales. We have provided empirical evidence that DRM can lead to strong generalization to covariate and concept shifts in the presence of spurious correlations in training data. We end with a Q&A discussion on limitations of DRM, potential ways to address them, and other exciting directions for future work. See Appendix E for additional discussion.

**Q: When does the DRM objective fail to lead to OOD generalization?** Broadly, there are three possible failure modes of DRM. First, it may be possible to find representations that make *training* data appear practically iid, but that do not lead to making the combination of training *and* test data practically iid. This can occur if the axes of variation seen in training data do not span differences between training and test data (e.g., in the imitation learning example from Sec. 5.3, the table and bowl colors were only varied along the blue and green channels, and thus will not lead to generalization when the red channel is altered). Care should be taken to curate training data that span as many relevant axes of variations as possible, even if the *magnitude* of variations is not representative of test data. The second failure mode is when the distribution shift detector we are deceiving is not sufficiently powerful. We expect that continued progress in distribution shift detection will lead to improvements in DRM. Another particularly promising direction is to simultaneously train both the data representation and the detector as an adversarial game. Third, there may be cases where it is not feasible to find representations that eliminate distribution shift in training data, but where one can find invariant predictors as advocated by IRM (Arjovsky et al., 2019, Appendix C). In such cases, DRM is not the right tool. We also note that Rosenfeld et al. (2020) construct data-generating distributions that cause IRM to fail. Since Rosenfeld et al. (2020) consider IRM and related objectives that find invariances across a finite number of data-generating distributions, the results are not directly applicable to DRM. An interesting theoretical direction is to characterize the precise conditions under which a DRM-style objective can lead to OOD generalization. We provide a preliminary sketch of theoretical underpinnings of DRM in Appendix F by connecting deception to generalization.

**Q: What are the computational challenges related to implementing DRM?** The primary computational bottleneck is in Eq. 3 and Eq. 4, which compute the conformity scores for each example. For each example in the sequence of data points used for distribution shift detection, we compute the minimum distance in embedding space to other examples in the sequence (quadratic complexity). Currently, we address this by sampling subsequences of data from the training sequence, and using these to compute martingale values which are then averaged (Appendix A). Finding strategies to improve this computational bottleneck — perhaps with inspiration from efficient implementations of the quadratic-complexity attention mechanism (Zhuang et al., 2023) — is an important avenue for making DRM scalable.

**Q: How sensitive is DRM to different hyperparameters?** The primary hyperparameters in DRM are: the dispersion parameter $\sigma$ for smooth sorting (Sec. 4), the length of the sequences used for distribution shift detection (Sec. A), and the relative weighting $\lambda$ between the ERM objective and the DRM regularization (Eq. 7). Hyperparameters chosen for the numerical experiments are reported in Appendix B, and we present results from a hyperparameter sweep for the 2D example in Appendix C. We find that DRM is sensitive to the dispersion parameter $\sigma$ and relatively insensitive to the weighting $\lambda$ and the length of the detection sequences.

**Q: Are there other kinds of distribution shift that could be handled by a DRM-style objective?** The two kinds of distribution shift we have considered in this paper are covariate shift and (anti-causal) concept shift. Chapter 8.2 of Vovk et al. (2022) presents a conformal martingale for detecting *label shift*, i.e., a shift in the marginal distribution of class labels. Causal concept shift— a change in the conditional distribution $Y|X$ — is highly relevant in causal inference. In the absence of label shift and covariate shift, anti-causal and causal concept shift are equivalent (via Bayes' rule), as in our examples from Sec. 5.1 and 5.2. Extending DRM to tackle causal concept shift in general settings is an important avenue for future work.

Overall, we are excited by the prospect that the bridge between distribution shift detection and generalization provided by DRM will lead to new techniques that address the problem of OOD generalization, which remains prevalent despite the scale of modern machine learning.

## REPRODUCIBILITY STATEMENT

The paper provides all algorithmic details (Sec. 4), implementation details (Appendix A), and hyperparameters (Appendix B) for reproducing results from experiments. In addition, code for all experiments is provided as part of the submission. Results can be reproduced with a single RTX 4090 GPU.

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

# A    ALGORITHMIC IMPLEMENTATION DETAILS

Algorithm 2 presents the steps for computing the soft martingale regularization for the DRM problem.

---

**Algorithm 2** Computing the DRM regularizer

---

1: **Input:** sequence of features $(\phi(x_t))_{t=1}^T$; choose shift type $s \in \{\text{covariate}, \text{concept}\}$
2: **Output:** sequence of soft martingale values $(\tilde{S}_t)_{t=1}^T$
3: **for** $t \leftarrow 1\ T$ **do**
4:     **for** $i \leftarrow 1\ t$ **do**
5:         **if** $s = \text{covariate}$ **then**
6:             $\tilde{\alpha}_i \leftarrow \text{soft}\big(\alpha_i^{\text{covariate}}\big)$                 $\triangleright$ replace min with soft-min in equation 3
7:         **else**
8:             $\tilde{\alpha}_i \leftarrow \text{soft}\big(\alpha_i^{\text{concept}}\big)$                 $\triangleright$ replace min with soft-min in equation 4
9:         **end if**
10:    **end for**
11:    **if** $s = \text{covariate}$ **then**
12:        $\tilde{p}_t^{\text{covariate}} \leftarrow \text{soft}\big(p_t^{\text{covariate}}\big)$             $\triangleright$ use equation 5 with soft-quantile
13:    **else**
14:        $\tilde{p}_t^{\text{concept}} \leftarrow \text{soft}\big(p_t^{\text{concept}}\big)$             $\triangleright$ use equation 6 with soft-quantile
15:    **end if**
16: **end for**
17: Compute $(\tilde{S}_t)_{t=1}^T$ using Algorithm 1 with inputs $(\tilde{p}_t^{\text{covariate}})_{t=1}^T$ or $(\tilde{p}_t^{\text{concept}})_{t=1}^T$.

---

We additionally discuss a few implementation details for DRM.

**Multiple detection sequences.** In settings where the training data sequence is large, we subsample multiple sequences, compute (soft) martingales for each, and average these to form the regularization term in Eq. 7. This results in improved computational efficiency and robustness compared to computing a single martingale value from the entire training data sequence.

**Feature normalization.** As described in Sec. 4, we utilize cosine distances to define conformity scores. Since the resulting conformity scores are only sensitive to directional differences between features, we normalize encodings to have unit norm, i.e., $\|\phi(x)\|_2 = 1$.

**Warm-starting with ERM.** For the Colored-MNIST example (Sec. 5.2), we found that warm-starting DRM with a small number of epochs of ERM helped improve performance. This is consistent with the implementation of invariant risk minimization (IRM) from Arjovsky et al. (2019).

# B    HYPERPARAMETERS FOR NUMERICAL EXPERIMENTS

| Parameter | Toy 2D Example | Colored-MNIST | Imitation Learning |
|---|---|---|---|
| # training examples (T) | 2000 | 2000 | 300 |
| ERM loss batch size | 64 | 64 | 64 |
| Size of detection sequence | 1000 | 1000 | 200 |
| # of detection sequences | 1 | 3 | 3 |
| Regularization weight ($\lambda$) | 5e5 | 5e6 | 1e4 |
| Dispersion for soft-ranking ($\sigma$) | 0.001 | 0.1 | 0.001 |
| Learning rate | 0.005 | 0.005 | 0.001 |
| # ERM epochs | 0 | 2 | 0 |
| # total training epochs | 2 | 3 | 25 |

## C Hyperparameter sweep for 2D example

The following table shows success rates for different values of the regularization weight ($\lambda$), averaged across 5 training seeds.

|  | $\lambda = 1e4$ | $\lambda = 1e5$ | $\lambda = 5e5$ | $\lambda = 1e6$ |
|---|---|---|---|---|
| Success (Train $\parallel$ Test) | 0.67 $\parallel$ 0.72 | 0.74 $\parallel$ 0.57 | 0.70 $\parallel$ 0.63 | 0.69 $\parallel$ 0.56 |

The following table shows success rates for different values of the detection sequence length, averaged across 5 training seeds.

|  | 200 | 400 | 600 | 800 | 1000 |
|---|---|---|---|---|---|
| Success (Train $\parallel$ Test) | 0.69 $\parallel$ 0.53 | 0.69 $\parallel$ 0.68 | 0.71 $\parallel$ 0.63 | 0.71 $\parallel$ 0.56 | 0.70 $\parallel$ 0.63 |

The following table shows success rates for different values of the soft-rank dispersion parameter ($\sigma$), averaged across 5 training seeds.

|  | $\sigma = 0.001$ | $\sigma = 0.01$ | $\sigma = 0.1$ |
|---|---|---|---|
| Success (Train $\parallel$ Test) | 0.70 $\parallel$ 0.63 | 0.65 $\parallel$ 0.56 | 0.71 $\parallel$ 0.51 |

## D Imitation learning example details

The task is to pick up and place a red block into a bowl using observations from an RGB camera. The training data consists of 300 expert demonstrations of pick-and-place locations, which are provided in different environments. A third of the demonstrations are provided with one table-and-bowl color combination (Table RGB: [0, 0.2, 0.7], Bowl RGB: [0, 0, 0.5]), , the next third with a slightly different combination (Table RGB: [0, 0.4, 0.9], Bowl RGB: [0, 0.2, 0.7]), and the final third with another combination (Table RGB: [0, 0.3, 0.6], Bowl RGB: [0, 0.6, 0.3]); these are visualized in Fig. 1. At test time, the bowl and table background color are changed to a novel combination (Table RGB: [0, 0.9, 0.4], Bowl RGB: [0, 0.7, 0.2]) that significantly exaggerates the variation in green and blue channels seen during training (Fig. 1 right).

## E Additional discussion

**Q: Can other methods be used for distribution shift detection in place of conformal test martingales?** In this work, we instantiated DRM using conformal martingales (CMs). This choice was motivated by (i) prior work that demonstrates the ability of CMs to detect distribution shifts rapidly (Vovk et al., 2022), (ii) the ability of CMs to detect different kinds of distribution shifts (e.g., covariate and concept shifts), and (iii) the fact that we can construct a differentiable surrogate for CMs. There is exciting future work in contrasting the theoretical and empirical benefits of utilizing other approaches to distribution shift detection (Sec. 6). An approach based on a different detector may make DRM more computationally efficient.

**Q: Could DRM be used for covariate shifts due to compounding errors in imitation learning?**
**A:** One idea is to implement the iterative data collection process in DAGGER (dataset aggregation) (Ross et al., 2011), and use DRM to find features that remain robust to the covariate shift between the states visited in successive iterations. Such a strategy may lead to more robust policies compared to DAGGER, which re-trains the policy by aggregating data across iterations of data collection. Working out the details of such an approach could make for interesting future work.

**Q: Could DRM be used for reinforcement learning?**
**A:** One immediate application of DRM in reinforcement learning (RL) is in the setting where one has access to a sequence of Markov decision processes (MDPs) for training (similar to the imitation learning setup considered in Sec. 5.3). In this case, the distribution shift detector can take as input observations from different environments, and the DRM objective would then attempt to learn a policy whose features appear iid across environments.

# F  DECEIVE TO GENERALIZE: THEORETICAL INTUITIONS

In this section, we draw theoretical connections between the objective of deceiving a distribution shift detector and that of achieving OOD generalization. The connection is made in three parts. First, we demonstrate that if a particular distribution shift detector $\Delta^\star$ can be deceived into concluding that the random variables corresponding to training and test losses are iid, then the expected test loss is very close to the expected training loss. Second, we allow for detectors that take encoded representations $\phi(x)$ as input instead of loss values. Third, we define the $\Delta$-span of a representation learned from training random variables as containing test distributions such that training and test random variables are practically iid. Any test distribution in the span thus has expected test loss close to the expected training loss.

## F.1  EFFICIENCY OF DISTRIBUTION SHIFT DETECTION

In Sec. 4, we defined observers $\Delta$ in the form of distribution shift detectors that control the false alarm rate (FAR). A detector should also ideally detect distribution shifts as quickly as possible. The notion of efficiency can be formalized by the worse average delay (WAD) of a detector.

**Worst average delay (WAD).** Suppose that the marginal distributions of the sequence of random variables $(\phi(X_1), \phi(X_2), \dots)$ change at an unknown time $\nu$, referred to as a *changepoint*. The worst average delay (WAD) in detecting the change is (Shin et al., 2022):

$$\sup_{\nu \geq 0} \mathbb{E}[N^\star - \nu | N^\star > \nu], \tag{9}$$

where $N^\star$ is the time at which a distribution shift is declared ($N^\star = \infty$ if a change is never declared).

The following definition formalizes the idea of random variables with a changepoint appearing iid to a given observer. Intuitively, the sequence of random variables is practically iid if the worst average delay in detecting a changepoint is large.

**Definition 2** (Practically iid w/ changepoint). *A sequence of random variables $(\phi(X_1), \dots, \phi(X_\nu), \phi(X_{\nu+1}), \dots)$ with changepoint $\nu$ is $(\Delta_\alpha, \epsilon)$-practically iid if the detector $\Delta_\alpha$ with FAR bounded by $\alpha$ has a large WAD in detecting the changepoint: WAD $> (1/\epsilon) \log(1/\alpha)$.*

## F.2  CONNECTING DETECTION TO GENERALIZATION

Consider the sequence of input random variables $(X_1, \dots, X_T, X_{T+1} \dots)$ as in Sec. 3, where the changepoint $T$ separates training and test distributions. We will demonstrate that there is an encoding $\phi$ of inputs and a detector $\Delta_\alpha^\star$ such that if $(\phi(X_1), \dots, \phi(X_T), \phi(X_{T+1}), \dots)$ is $(\Delta_\alpha^\star, \epsilon)$-practically iid, then the expected test loss is close to the expected training loss.

**Proposition 1.** *Let $h$ be a hypothesis that maps inputs to labels, and consider a binary-valued loss function, i.e., $l(x, h(x)) \in \{0, 1\}, \forall x$. Suppose that the expected loss under the training random variables is bounded as follows:*

$$\mathbb{E}[l(X_t, h(X_t))|\mathcal{F}_{t-1}] \leq l_{train}, \ \forall t \leq T, \tag{10}$$

*where $\mathcal{F}_{t-1}$ denotes the natural filtration of the data. Consider the sequence of random variables $(X_1, \dots, X_T, \dots)$, where the test random variables $(X_{T+1}, X_{T+2}, \dots)$ are iid. Then the expected test loss is:*

$$l_{test} := \mathbb{E}[l(X_{T+1}, h(X_{T+1}))|\mathcal{F}_T]. \tag{11}$$

*There exists a detector $\Delta_\alpha^\star$, an encoding function $\phi$, and a constant $c$ such that the following result holds in the limit as $\alpha \to 0$. If $(\phi(X_1), \dots, \phi(X_T), \phi(X_{T+1}), \dots)$ are $(\Delta_\alpha^\star, \epsilon)$-practically iid, then:*

$$kl(l_{test}\|l_{train}) \leq c\epsilon, \tag{12}$$

*where $kl(\cdot\|\cdot)$ is the KL-divergence between two Bernoulli random variables with parameters $l_{test}$ and $l_{train}$.*

*Proof.* Define $\phi : x \mapsto l(x, h(x))$. The random variables $(\phi(X_1), \dots, \phi(X_{T+1}), \dots)$ then correspond to Bernoulli random variables with dependent, time-varying means. In the limit

$\alpha \to 0$, the detector presented by Shin et al. (2022) has FAR bounded by $\alpha$ and achieves a WAD $\leq c \log(1/\alpha)/\mathrm{kl}(l_{\text{test}} \| l_{\text{train}}))$. Now, suppose for contradiction that $\mathrm{kl}(l_{\text{test}} \| l_{\text{train}}) > c\epsilon$. Then, we have WAD $< (1/\epsilon) \log(1/\alpha)$, which contradicts the statement that $(\phi(X_1), \dots, \phi(X_T), \dots)$ are $(\Delta_\alpha^\star, \epsilon)$-practcially iid. $\square$

The practical utility of the detector $\Delta_\alpha^\star$ above is limited since it takes losses as input; because we ultimately rely only on making training data practically iid, $\Delta_\alpha^\star$ can be deceived into not detecting a distribution shift on training data simply by overfitting and driving the loss on all training examples to 0. To address this challenge, we allow for detectors (e.g., based on conformal martingales) that take latent representations $\phi(x) \in \mathbb{R}^d$ as input. The following corollary follows immediately from Proposition 1.

**Corollary 1.** *Let $h_\phi$ be a hypothesis with latent encoding $\phi$. Consider a detector $\Delta_\alpha$ that observes inputs encoded by $\phi$, and that is at least as efficient as the detector $\Delta_\alpha^\star$ that relies on loss values, i.e., the WAD of $\Delta_\alpha$ for any pre- and post-change distributions is less than or equal to the WAD of the detector $\Delta_\alpha^\star$. Then, there exists a constant $c$ such that the following result holds in the limit as $\alpha \to 0$. If $(\phi(X_1), \dots, \phi(X_T), \phi(X_{T+1}), \dots)$ are $(\Delta_\alpha, \epsilon)$-practically iid, then $\mathrm{kl}(l_{test} \| l_{train}) \leq c\epsilon$.*

The results above rely on having access to test data. Instead, consider a representation $\phi$ such that the training data sequence $(\phi(x_1), \dots, \phi(x_T))$ is $\Delta$-practically iid, and define the $\Delta$-span of this representation as containing test distributions such that $(\phi(X_1), \dots, \phi(X_T), \phi(X_{T+1}), \dots)$ are $(\Delta_\alpha, \epsilon)$-practically iid. Then, in the limit as $\alpha \to 0$, it follows from the results above that for any test distribution in the $\Delta$-span, $\mathrm{kl}(l_{\text{test}} \| l_{\text{train}}) \leq c\epsilon$.