# OpenReview forum: "Deceptive Risk Minimization: Out-of-Distribution Generalization by Deceiving Distribution Shift Detectors"
_ICLR.cc/2026/Conference — ICLR 2026 Conference Withdrawn Submission_

### Official Review · Reviewer_VS2s · 2025-10-20

**Soundness:** 2
**Presentation:** 3
**Contribution:** 2
**Rating:** 2
**Confidence:** 4

**Summary:**

This paper proposes DRM that learns representations to fool a sequential distribution-shift detector to judge the ordered training sequence as i.i.d., thereby achieving OOD generalization. Algorithmically, DRM is implemented as a differentiable martingale penalty by softening the conformal steps and backpropagating through the martingale updates. The experiments include a 2D concept-shift toy dataset, ColorMNIST, and an imitation-learning covariate-shift setting, where DRM shows decent qualitative performance.

**Strengths:**

* The proposed method bridges OOD detection and generalization, providing a new perspective on how to achieve continuous domain generalization. The framework is intuitive and insightful.
* The experimental results show clear improvement over the ERM baseline.

**Weaknesses:**

* While the studied problem is new, the current implementation is based on conformal martingale, which is already well-established in previous papers. The softening technique is also borrowed from previous works, so there's limited contribution on the methods side.
* The time complexity of the current implementation is $O(T^2)$, which would be high compared to conventional domain generalization methods. The paper does not include comparisons of running time, so it's unclear how efficient the proposed method is.
* The reported results are all conducted on simple toy datasets, and no quantitative performance comparisons are included. There is also a lack of comparison to baseline methods for continuous domain generalization, e.g., [1-2]. The performance improvement of the proposed method is not convincing based on the current results.

[1] Generalizing to Evolving Domains with Latent Structure-Aware Sequential Autoencoder, 2022.

[2] Enhancing Evolving Domain Generalization through Dynamic Latent Representations, 2024.

**Questions:**

* I would suggest that the authors add more extensive experimental results on more real-world datasets and compare to the missing baseline methods (as above) to enhance the paper.

---

### Official Review · Reviewer_GH2j · 2025-10-31

**Soundness:** 2
**Presentation:** 3
**Contribution:** 3
**Rating:** 4
**Confidence:** 3

**Summary:**

To learn invariant features given non-i.i.d. training data, the paper proposes deceptive risk minimization (DRM), which encourages the representations of the non-i.i.d. training data to be viewed as i.i.d. by a distribution shift detector. DRM regularizes the learning with a loss function that reduces the betting conformal martingales of the distribution shift detector. Experiment on covariate shift and concept shift toy datasets shows that the proposed method works well in learning invariant features.

**Strengths:**

1. The motivation of DRM is intuitive and reasonable, and using a conformal martingale as an objective is novel.

2. DRM does not require domain labels. It has fewer constraints and is more generalizable.

3. Figures 4 and 5 are illustrative and very helpful in showing that DRM learns representations that contain little spurious features.

**Weaknesses:**

The empirical study is not convincing enough in showing the robustness and the superiority of the method. Specifically,
1. How robust is the distribution shift detector with high-dimensional representation learned by modern networks such as ResNet?

2. What is the performance of DRM on real-world datasets (e.g., waterbirds, CelebA, MultiNLI, Wilds benchmark, DomainBed benchmark etc.) where spurious correlation/covariate shift is more complex, beyond the color shown in ColoredMNIST and the robot imitation learning example?

3. How is the performance of DRM compared with other SOTA OOD methods that learn invariant features? IRM was relatively rudimentary, and there are a number of methods that improve upon it.

**Questions:**

1. The ordering requirement in computing the conformal p-value incurs a high computational cost. Why does ordering matter in equations 5 and 6? Can pairwise non-ordered conformity score be used in the calculation?

---

### Official Review · Reviewer_6Wow · 2025-11-07

**Soundness:** 2
**Presentation:** 1
**Contribution:** 2
**Rating:** 2
**Confidence:** 3

**Summary:**

This paper focuses on mitigating domain shifts and enhancing OOD generalization. Authors propose deception as a new mechanism that learns representations for making training data appear independent to an observer. Authors refer to this principle as deceptive risk minimization (DRM) and leverage the idea of conformal martingales. Experiments on small-scale datasets, including Toy 2D Examples, Colored-MNIST, and Imitation Learning, validate the efficacy of DRM.

**Strengths:**

1. Reframing OOD generalization as a problem of "deceiving" a sequential shift detector seems to be a unique perspective.
2. The connection between distribution shifts and robot behaviors increases the potential practical utility of the proposed method.

**Weaknesses:**

1. The introduction seems to be overly complex. Several concepts are hard to understand. For example, domain shift usually describes inherent distribution discrepancy; why can it be mitigated from "the perspective of an external observer"? How can the external observer eliminate the distribution shift?  Moreover, the core challenge that this paper is trying to solve is not very clear.
2. The paper is built on a strong assumption that all data are sequential, which represents a considerably simplified setting compared to real-world scenarios, where data are typically randomly permuted and non-sequential.
3. This paper explores specific training data that reflect "some (possibly mild)" distribution shifts. However, the description is vague and lacks quantitative characterization. How is the degree of "mild" distribution shift defined and measured?
4. Since the core Conformal Martingale Detector has quadratic complexity, it is inherently computationally expensive. This limitation hinders the extension of this method to real large-scale datasets.
5. The main experiments are conducted on small-scale datasets (2000 training examples for Toy 2D Examples and Colored-MNIST, 300 training examples for Imitation Learning), lacking evaluation on large-scale real-world datasets like DomainBed. Additional experiments on such large-scale datasets would strengthen the validation of the proposed method.
6. The evaluation on Colored MNIST is restricted to a binary classification task involving only two color variations (red and green). This narrow experimental setup limits the generalizability of the findings and does not convincingly validate the method’s applicability to real-world scenarios.

**Questions:**

Please see the weakness.

---

### Official Review · Reviewer_6yoo · 2025-11-11

**Soundness:** 2
**Presentation:** 3
**Contribution:** 2
**Rating:** 4
**Confidence:** 3

**Summary:**

This paper introduces Deceptive Risk Minimization (DRM), a novel framework for out-of-distribution (OOD) generalization that trains models to “deceive” distribution shift detectors by making data representations appear independent and identically distributed (iid). The method integrates conformal martingales into a differentiable objective, enabling the learning of stable, shift-invariant features without requiring predefined domains or test data. Empirical results on toy datasets, Colored-MNIST, and robotic imitation learning show promising robustness compared to empirical and invariant risk minimization baselines.

**Strengths:**

1. Conceptual Innovation: Proposes a novel deceptive perspective on out-of-distribution (OOD) generalization, framing robustness as deceiving distribution shift detectors — an original and conceptually intriguing idea.

2. Practical Flexibility: Unlike prior invariant risk minimization (IRM) or domain adaptation methods, DRM does not require predefined domains or access to test data, making it applicable to continuously shifting real-world environments.

3. Empirical Performance: Demonstrates strong robustness to covariate and concept shifts in toy, image, and robotic tasks, achieving IRM-level generalization without oracle knowledge of domain boundaries.

**Weaknesses:**

1. Dependence on Shift Detector: DRM’s success is tightly coupled to the capability of the distribution shift detector; weak or poorly tuned detectors may fail to identify spurious correlations, limiting its robustness.

2. Computational Overhead: The conformal martingale regularization introduces quadratic computational complexity with respect to data sequence length, which may hinder scalability to large or high-dimensional datasets.

3. Sequential Data Assumption: The framework assumes that data are temporally ordered and exhibit gradual distribution shifts, a condition rarely satisfied in static or shuffled datasets.

4.  **Unrealistic Domain Shift Assumption**: DRM implicitly assumes that domain shifts **can be clearly simulated or defined**. However, in real-world scenarios, such *domain boundaries are often ill-defined or unobservable*. If domain shifts were truly definable, similar generalization could be achieved through data generation, augmentation, or standard domain adaptation methods—making this assumption somewhat idealized. And this is a **Lack of Evaluation on Standard Benchmarks**.  The experiments are limited to *small-scale or synthetic datasets*. Validation on domain generalization benchmarks such as DomainBed, PACS, or Office-Home would significantly strengthen empirical credibility.

5. Limited Theoretical Grounding: The theoretical justification connecting “deception” to OOD generalization remains preliminary and lacks rigorous formal proofs or convergence analysis.

**Questions:**

N/A

---

### Note · Authors · 2025-11-13

I have read and agree with the venue's withdrawal policy on behalf of myself and my co-authors.